Diversity and habitat preferences of bdelloid rotifers in mosses and liverworts from beach forest along sand dunes in Thailand

Jattupan Sittikron 1
Jaturapruek Rapeepan 1
Sa-ardrit Phannee 2
Inuthai Janejaree 3
Ngernsaengsaruay Chatchai 4 5
Maiphae Supiyanit 1 5 supiyanit.m@ku.th
1 Animal Systematics and Ecology Research Unit, Department of Zoology, Faculty of Science, Kasetsart University , Chatuchak, Bangkok , Thailand
2 The Princess Maha Chakri Sirindhorn Natural History Museum and RSPG Southern Region Network Coordinating Center, Prince of Songkla University , Hatyai, Songkhla , Thailand
3 Department of Biotechnology, Faculty of Science and Technology, Thammasat University, Lampang Campus , Hang Chat, Lampang , Thailand
4 Department of Botany, Faculty of Science, Kasetsart University , Chatuchak, Bangkok , Thailand
5 Biodiversity Center Kasetsart University, Kasetsart University , Chatuchak, Bangkok , Thailand
Roper James
Electronic publication date: 2024 Dec 16
Publication date: 2024
Volume: 12
Electronic Location ID: e18721
Received 2024 Jul 31; Accepted 2024 Nov 26
Copyright: © 2024 Jattupan et al.
Copyright year: 2024
Copyright holder: Jattupan et al.
License: This is an open access article distributed under the terms of the Creative Commons Attribution License, which permits unrestricted use, distribution, reproduction and adaptation in any medium and for any purpose provided that it is properly attributed. For attribution, the original author(s), title, publication source (PeerJ) and either DOI or URL of the article must be cited.
License URL: https://creativecommons.org/licenses/by/4.0/

Keywords: Biodiversity, Bryophytes, Distribution, Desiccation, Specificity, Ecological association, Microhabitats, Beach forest, Harsh environments, Species richness

Funding: Graduate Program Scholarship from The Graduate School, Kasetsart University International SciKU Branding (ISB), Faculty of Science, Kasetsart University This project was supported by the Graduate Program Scholarship from The Graduate School, Kasetsart University and the International SciKU Branding (ISB), Faculty of Science, Kasetsart University. The funders had no role in study design, data collection and analysis, decision to publish, or preparation of the manuscript.

==============================
Microscopic animals are often thought to be widely distributed due to their small size and specific adaptations. However, evidences show that bdelloid rotifers in bryophytes exhibit habitat specialization, with species composition varying by microhabitat. This indicates that their distribution is influenced by complex ecological processes, warranting further research, particularly at the microscale. In this study, we tested whether species richness and composition of bdelloid rotifers differ across bryophyte species, forms, characteristics, and seasons to understand their ecological distribution and habitat preferences in limnoterrestrial environments. Bdelloid rotifers were identified and counted from bryophyte samples collected in April (low rainfall), August (moderate rainfall), and December 2022 (high rainfall) at Bang Burd Beach Forest, Chumphon Province, Thailand. The results revealed high bdelloid diversity, with 22 species identified, 14 of which are new records for Thailand, raising the known number to 30. However, species richness did not vary across bryophyte variables or seasons, with substantial overlap in species composition across these variables. Additionally, there was no strong habitat preference between bdelloid rotifers and bryophyte species. These results confirm a low degree of habitat specialization of bdelloid rotifers in tropical limnoterrestrial environments.

Introduction

The distributions and abundances of microscopic organisms (<1 mm) may not follow the patterns typically attributed to larger organisms (Fontaneto et al., 2008; Segers & De Smet, 2008; Kaya & Erdoğan, 2015; Zawierucha et al., 2023). It has been hypothesized that microscopic organisms tend to be more widely distributed because they are small enough to be passively dispersed by wind over long distances, a concept known as the ‘everything is everywhere’ or ubiquity hypothesis (Finlay, 2002; Fenchel & Finlay, 2004; Fontaneto & Hortal, 2013). Additionally, certain microscopic organisms, such as rotifers, possess efficient resting stages that allow them to endure prolonged periods of dormancy, and their asexual and parthenogenetic reproduction enables them to rapidly colonize suitable habitats (Fontaneto, 2019). This suggests that they can be considered cosmopolitan (Fontaneto, 2011). Furthermore, Fontaneto, Westberg & Hortal (2011) confirmed that microscopic organisms, such as bdelloid rotifers, have a lower degree of habitat specialization than larger organisms. However, this occurs within a complex scenario of ecological processes; therefore, more research is needed to explain the effects on species composition, especially at the microscale.

The distribution of microinvertebrates, such as tardigrades (Nelson & Adkins, 2001; Ramsay et al., 2021) and bdelloid rotifers (Kaya, De Smet & Fontaneto, 2010; Dražina et al., 2013; Kaya & Erdoğan, 2015), in bryophytes has been extensively studied. Bryophytes have ecological associations with microorganisms, including protozoans and rotifers, as well as other invertebrates, plants, and fungi (Gerson, 1982). They provide food, shelter, and nesting material for small animals and invertebrates, indirectly serving as a matrix for various interactions among all these organisms (Bahuguna et al., 2013). Several studies have illustrated significantly enhanced invertebrate densities in bryophytes compared to unstable gravels (Suren, 1991, 1993). Furthermore, it has been reported that the species richness and composition of bdelloid rotifers living in bryophytes differ significantly among microhabitats, providing evidence of habitat specialization (Kaya & Erdoğan, 2015; Heatwole & Miller, 2019). In contrast, no relationships have been found between bdelloids and moss species (Burger, 1948; Kaya, De Smet & Fontaneto, 2010). Therefore, it appears that the species composition inhabiting bryophytes may change over time, or certain species may coexist in specific patterns.

Bdelloid rotifers are microscopic organisms capable of reproducing without fertilization and resisting dry and frozen conditions, which allows them to disperse across a variety of terrestrial and aquatic habitats (Fontaneto, Melone & Ricci, 2003; Ricci & Caprioli, 2005; Fontaneto & Ricci, 2006; Wilson, 2011; Debortoli, Laender & Doninck, 2018). Currently, approximately 460 species of bdelloid rotifers have been reported worldwide (Segers, 2007). In recent decades, bdelloid diversity in terrestrial environments has been extensively studied across various regions, including Central and Eastern Europe (Donner, 1965; Koste, 1975, 1978a, 1978b; Schmid-Araya, 1995), Turkey (Kaya, 2013), Korea (Song & Kim, 2000; Song & Min, 2015; Song & Lee, 2017), China (Zeng et al., 2020; Wang et al., 2023), the Arctic (Svalbard) (Kaya, De Smet & Fontaneto, 2010), and Antarctica (Velasco-Castrillón et al., 2014b; Iakovenko et al., 2015). These studies have contributed to extensive species lists, resulting in numerous species being recorded (Fontaneto et al., 2007; Zeng et al., 2020).

In Thailand, studies on bdelloid rotifers have predominantly focused on freshwater habitats, with 16 species reported (Sa-Ardrit, Pholpunthin & Segers, 2013; Maiphae, 2017; Jaturapruek et al., 2018, 2021). While their niche preferences and distribution patterns in freshwater environments have been explored (Jaturapruek et al., 2021), there remains limited knowledge about their distribution in limnoterrestrial habitats. This study was the first to investigate the habitat preferences of bdelloid rotifers in tropical terrestrial environments, where water availability may influence dispersal and distribution patterns, potentially differing from those in drier or colder regions. The Bang Burd Forest, the largest beach forest on sand dunes in Thailand, is a unique ecosystem characterized by a diverse plant community, including climbers and trees adapted to sandy soils, intense sunlight, high winds, dryness, and constant seawater spray (Inuthai, 2007). This forest is significant not only for its distinct environmental conditions but also for the rich biodiversity it supports. Previous research has documented a high diversity of bryophytes—specifically, 16 species of mosses and liverworts (Inuthai, 2007)—indicating a variety of microhabitats that could be crucial for supporting invertebrate communities (Budke et al., 2018). As a result, this study aims to examine the distribution patterns of bdelloid rotifers in relation to bryophyte species and their characteristics. By understanding these relationships, we hope to gain insights into the habitat preferences of bdelloid rotifers in this distinctive ecosystem.

Materials and Methods

Sampling site

Bang Burd Beach Forest in Chumphon Province, Thailand, is located near the coast (10°59′15.1″N 99°29′44.6″E) with an elevation ranging from 6 to 54 m above sea level. This forest is characterized by sand dunes and scattered patches of plants. The surrounding area is dry and sunny, while the interior is more humid. The plant community includes shrubs, trees, and climbers, which provide habitats for bryophytes (Fig. 1).

Figure 1 Sampling site.

(A) Sampling site at Bang Burd Beach Forest, Chumphon Province, Thailand and (B, C) Bang Burd Beach Forest environment area.

Sample collection, species identification and count

A total of 173 bryophyte samples were collected in April 2022 (low rainfall), August 2022 (moderate rainfall), and December 2022 (high rainfall). The average rainfall for each period, based on 2019–2021 data, was 6.96, 15.04, and 21.64 mm, respectively (Hydro-Informatics Institute (Public Organization), 2021). All samples were stored in zip-lock plastic bags for bdelloid studies and bryophyte identification. These samplings were approved by the Institutional Animal Care and Use Committee, Kasetsart University (approval no. ACKU66-SCI-018).

There are 11 species of mosses and 31 species of liverworts which were classified into groups (moss, liverwort), life forms (cushion, turf, mat) (Inuthai, 2007; Suwanmala & Chantanaorrapint, 2016), and morphological characteristics. The morphological characteristics of moss and liverwort were classified by the complexity of their structure that may be linked to the ability to harbor water, which is an environment for rotifers. Mosses are classified into two characters including leaves curl when dry and leaves do not curl when dry (in this context, ‘leaves’ refers to a leave-like structure or phyllids) and liverworts are classified into two characters including large lobules (a ratio of lobules is about half or more than half of lobe length) and small lobules (a ratio of lobules is less than half of lobe length) (Table 1, Figs. 2–3). Samples containing mixed bryophyte species (Table S1) and those where bdelloid rotifers were not found were excluded from the analysis. Therefore, a total of 52 samples were used for data analysis. In the laboratory, samples were prepared for identification using a modified method from Peters, Koste & Westheide (1993). A 3 × 3 cm2 of each sample was soaked in mineral water for 24 h, then shaken thoroughly before collecting the water samples for bdelloid rotifer identification and counting. Bdelloid rotifers in each water sample were sorted using a stereomicroscope (Olympus SZ51). The morphological characteristics of each specimen were examined live with a light microscope (Olympus CH2). All taxonomic characters were photographed and recorded on video. Identifications were based on morphological characteristics, following Donner (1965) and Ricci & Melone (2000).

Table 1 Bryophyte species that were used for data analysis.

Bryophyte species	Codes	Groups	Seasons	Life forms	Characters	
Acrolejeunea recurvata Gradst.	AR	Liverworts	Low rainfall	Mat	Large lobules	
Cheilolejeunea ceylanica (Gottsche) R. M. Schust. & Kachroo	CC	Liverworts	Low rainfall	Mat	Large lobules	
Cheilolejeunea cf. intertexa	CI	Liverworts	Low rainfall	Mat	Large lobules	
Cololejeunea planissima (Mitt.) Abeyw.	CP	Liverworts	Low, moderate, high rainfall	Mat	Large lobules	
Frullania ericoides (Nees) Mont.	FE	Liverworts	Moderate rainfall	Mat	Large lobules	
Lejeunea adpressa Nees	LA	Liverworts	Low, moderate, high rainfall	Mat	Small lobules	
Lejeunea cocoes Mitt.	LC	Liverworts	High rainfall	Mat	Small lobules	
Microlejeunea punctiformis (Taylor) Steph.	MP	Liverworts	Low, moderate, high rainfall	Mat	Large lobules	
Schiffneriolejeunea cumingiana (Mont.) Gradst.	SC	Liverworts	Low rainfall	Mat	Small lobules	
Schiffneriolejeunea tumida var. haskarliana (Gottsche) Gradst. & Terken	ST	Liverworts	Low, high rainfall	Mat	Large lobules	
Brachymenium sp.	Bsp	Mosses	Moderate rainfall	Cushion	Leaves curl when dry	
Calymperes erosum Müll. Hal.	CE	Mosses	Moderate, high rainfall	Turf	Leaves curl when dry	
Calymperes tenerum Müll. Hal.	CT	Mosses	Low, moderate rainfall	Turf	Leaves curl when dry	
Octoblepharum benitotanii Salazar Allen & Chantanaorr.	OB	Mosses	Moderate, high rainfall	Turf	Leaves do not curl when dry	
Octoblepharum poscii Magill & B. H. Allen	OP	Mosses	Moderate rainfall	Turf	Leaves do not curl when dry	
Taxithelium instratum (Brid.) Broth.	TI	Mosses	Moderate, high rainfall	Mat	Leaves do not curl when dry	

Figure 2 Bryophyte life forms.

(A) Cushion (Brachymenium sp.). (B) Turf (Calymperes erosum). (C) Mat (Frullania ericoides).

Figure 3 Bryophyte characters.

(A) Leaves curl when dry (Taxithelium instratum). (B) Leaves do not curl when dry (Octoblepharum benitotanii). (C) Large lobule (Schiffneriolejeunea tumida var. haskarliana). (D) Small lobule (Lejeunea adpressa). Red circles indicate lobules.

Data analysis

Species diversity

Shannon diversity and species evenness index was used to describe the species diversity of bdelloid rotifers among bryophyte species, among seasons (low, moderate and high rainfall), among bryophyte forms (cushion, turf, mat) and among bryophyte characters (mosses: leaves curl when dry and leaves do not curl when dry; liverworts: large lobules and small lobules). These analyses used the Microsoft Excel program (Microsoft 365).

Species composition and habitat preferences

Differences in species composition were assessed using a Jaccard similarity index (Wolda, 1981). Principal coordinates analysis (PCoA) was employed to visualize variations in the bdelloid community between samples, based on dissimilarity distances. This analysis was conducted using the PC-ORD program, version 7.11 (McCune & Mefford, 2016). Prior to analysis, the abundance of each species was log-transformed to normalize the data. It was conducted using a Euclidean as a distance measure. Convex hull polygons were used to highlight groups of variables including bryophyte group, bryophyte species, bryophyte form, bryophyte characters and rainfall. The habitat preference of each species was calculated using the equation proposed by Dufrêne & Legendre (1997). Based on the results, species were classified into three groups of preference degree: high (>70%), moderate (40–70%), and low (<40%).

Results

Species diversity

A total of 22 bdelloid species were identified (Table 2), 14 of which are newly recorded in Thailand. In addition, five bdelloid species, including Adineta vaga, Adineta sp. 2, Habrotrocha cf. brocklehursti, Macrotrachela cf. plicata, and Philodina rugosa, were found exclusively in samples of mixed bryophyte species (Table 2), making it difficult to determine the exact bryophyte species they inhabited. In each example, a range of 1 to 7 bdelloid species was identified, which corresponds to the number of species observed in other variables, including the bryophyte group, form, morphological characteristics, and across different seasons.

Table 2 Bdelloid rotifer species found in the present study and their distribution in mosses, liverworts and mixed bryophyte species.

Bryophyte abbreviations present in Table 1.* = new records in Thailand, + = present.

Bdelloid rotifer species	Mosses	Liverworts	Mixed bryophyte species	
*Adineta cf. glauca Murray, 1911	–	CI	–	
*A. vaga (Davis, 1873)	–	–	+	
Adineta sp.1	CT	–	–	
Adineta sp.2	–	–	+	
Didymodactylos sp.	–	MP	–	
Habrotrocha angusticollis (Murray, 1905)	CE, OB, TI	CP, FE, LA, MP, ST	+	
*H. cf. alacris Milne, 1916	–	ST	–	
*H. bidens (Gosse, 1851)	CE, CT, OB	CI, CP, LA, MP, ST	+	
*H. cf. brocklehursti Murray, 1911	–	–	+	
*H. flaviformis De Koning, 1947	–	CP, LA, ST	+	
*H. gracilis Montet, 1915	–	CC, CI	–	
Habrotrocha sp.	CE	–	–	
*Macrotrachela multispinosa Thompson, 1892	CE, CT, OB, OP, TI	CC, CP, FE, LA, LC, MP, ST	+	
*M. papillosa Thompson, 1892	–	CC, CP, MP, ST	+	
M. pinnigera (Murray, 1908)	Bsp	AR, LA	+	
*M. cf. plicata (Bryce, 1892)	–	–	+	
Macrotrachela sp.	OB	–	–	
*Philodina rugosa Bryce, 1903	–	–	+	
*P. verrucosa Song & Lee, 2020	CT	ST	+	
Pleuretra sp.	Bsp	CP, LA, LC	+	
*Rotaria sordida (Western, 1893)	Bsp, CT, OP, TI	CC, AR, CI, CP, FE, LA, LC, MP, SC	+	
*Scepanotrocha simplex De Koning, 1947	–	CC, LA, MP, ST	–	

Moreover, only two species, Macrotrachela multispinosa and Rotaria sordida, were distributed in more than 50% of the 83 samples, accounting for about 52% and 53%, respectively. They were followed by Habrotrocha angusticollis (25%) and Habrotrocha bidens (19%), which were relatively widespread. In contrast, most other species were found in only one to eight samples (Fig. 4).

Figure 4 The number of bryophyte samples found for each bdelloid rotifer species.

The species richness of bdelloid rotifers across different bryophyte species ranged from 1 to 8. Lejeunea adpressa and Schiffneriolejeunea tumida var. haskarliana contained the most diverse bdelloid rotifer species (eight species), followed by Cololejeunea planissima and Microlejeunea punctiformis (seven species), although the diversity index of Lejeunea adpressa was highest, followed by Cololejeunea planissima (Table 3).

Table 3 Shannon diversity of bdelloid rotifer species that are found in each bryophyte species.

Bold number indicated the highest value.

Bryophyte species	Species richness	Shannon diversity index	Evenness	
Acrolejeunea recurvata	2	0.33	0.47	
Brachymenium sp.	3	1.03	0.94	
Calymperes erosum	4	0.87	0.63	
C. tenerum	5	1.00	0.62	
Cheilolejeunea ceylanica	5	1.49	0.93	
C. cf. intertexa	4	1.21	0.87	
Cololejeunea planissima	7	1.58	0.81	
Frullania ericoides	3	0.85	0.77	
Lejeunea adpressa	8	1.73	0.83	
L. cocoes	3	1.04	0.95	
Microlejeunea punctiformis	7	1.40	0.72	
Octoblepharum benitotanii	4	0.95	0.69	
O. poscii	2	0.45	0.65	
Schiffneriolejeunea cumingiana	1	0	0	
S. tumida var. haskarliana	8	1.54	0.74	
Taxithelium instratum	3	0.72	0.66	

In addition, the bdelloid rotifer species richness found in liverworts (14 species) was higher than in mosses (10 species), which is in agreement with the trend in the diversity index (Table 4). Moreover, the highest species richness was observed in bryophytes with a mat life form, which supported 14 species, followed by turf with eight species and cushion with three species. This finding largely aligns with the trends observed in the diversity index, with the exception of the cushion life form, which exhibited the highest evenness value. Additionally, large-lobule liverworts displayed the highest richness, also at 14 species. Similarly, mosses with leaves that curl when dry supported nine species and five species were found in mosses with leaves that do not curl when dry. Notably, the highest diversity index was recorded for large-lobule bryophytes (1.93), followed by small-lobule bryophytes (1.68) (Table 4). Furthermore, the low rainfall period had the highest species richness and diversity index, with 14 species and a diversity index of 1.85. This was followed by the high rainfall period with nine species and a diversity index of 1.67, and the moderate rainfall period with eight species and a diversity index of 1.52 (Table 4).

Table 4 Diversity index of bdelloid rotifer species in each group, life forms, characters and seasons.

Bold number indicated the highest value.

Bryophytes	Species richness	Shannon diversity index	Evenness	
Groups				
Mosses	10	1.61	0.70	
Liverworts	14	1.97	0.75	
Life forms				
Cushion	3	1.03	0.94	
Mat	14	1.87	0.71	
Turf	8	1.54	0.74	
Characters				
Leaves curl when dry	9	1.67	0.76	
Leaves do not curl when dry	5	1.22	0.76	
Large lobule	14	1.93	0.73	
Small lobule	8	1.68	0.81	
Seasons				
Low rainfall	14	1.85	0.70	
Moderate rainfall	8	1.52	0.73	
High rainfall	9	1.67	0.76	

Notes on taxonomy of some unidentified bdelloid rotifers

Of the 22 recorded taxa, six species remain unidentified: Adineta sp.1, Adineta sp.2, Habrotrocha sp., Macrotrachela sp., Didymodactylos sp., and Pleuretra sp. (Figs. 5A–5N) due to insufficiently detailed characteristics. Additionally, some of these species exhibit characters that do not align with other members of their genera. Adineta sp.1 is notably small, measuring approximately 74.04 μm (Fig. 5A), while Adineta sp.2 is larger, measuring 176.92–197.12 μm (Fig. 5B), although it is still smaller than A. vaga (200–700 μm, according to Donner, 1965). Adineta sp.2 has an oval head that is wider than its trunk, but the rake apparatus is not clearly visible (Fig. 5C), and its foot is relatively short. These distinctive characteristics suggest that both Adineta sp.1 and Adineta sp.2 differ significantly from A. vaga. Additionally, the morphology of Didymodactylos sp. differs from the only known species in the genus, Didymodactylos carnosus (Figs. 5D–5F), notably lacking the bulbous base on the spurs characteristic of D. carnosus (Fig. 5E). Habrotrocha sp. has a shuttle-shaped body, with a neck nearly half the body length (Fig. 5G). Small pellets inside the stomach are visible. The spurs are short with three toes (Fig. 5H), and the trophi are relatively large with 3/4 teeth. In Macrotrachela sp., it is unclear whether the body is fully extended, as the first segment of the trunk is swollen (Fig. 5I). The spurs are short with three toes (Fig. 5J), and the trophi are small, with 2/2 teeth. Pleuretra sp. exhibits distinct morphological differences from known species and is considered a putative new species. It has a short head and neck, with a stiff, sculptured trunk covered in broad, blunt spines (Fig. 5K). The first trunk segment shows triangular processes in dorsal view (Fig. 5L). The foot and spurs are short, with four toes (Figs. 5M, 5N), and the trophi are small and symmetrical, with 3/3 teeth.

Figure 5 Photo of some bdelloid rotifers.

Adineta sp.1 (A) Feeding, dorsal view. Adineta sp.2 (B) creeping, dorsal view; (C) creeping head, dorsal view. Didymodactylos sp. (D) Creeping, dorsal view; (E) foot and spurs, lateral view; (F) toes, ventral view. Habrotrocha sp. (G) creeping, dorsal view; (H) foot and spurs, dorsal view. Macrotrachela sp. (I) creeping, dorsal view; (J) foot and spurs, dorsal view. Pleuretra sp. (K) trunk, lateral view; (L) trunk, dorsal view; (M) foot and spurs, dorsal view; (N) toes, ventral view (scale bars: B, E, F, H, I, M, N = 10 µm; A, C, D, G, J, K, L = 50 µm). The explanation of the arrows is given in the text.

Bdelloid rotifer community in bryophytes

The PCoA results indicate a high similarity among bdelloid species across each variable (Figs. 6 and 7). The first two PCoA axes for bryophyte species, groups, life forms, and rainfall accounted for a high proportion of the total variance (51.92%), with 31.37% explained by the first axis and 20.55% by the second axis. This reveals an overlap among the groups. These results are consistent with the Jaccard similarity index, which shows similarity levels of approximately 40–60% among bryophyte groups, bryophyte characteristics and seasons (Tables S2, S5, S6). Although most species are widely distributed, certain species such as Macrotrachela sp., Habrotrocha sp., and H. cf. alacris tend to be restricted to moss or liverwort habitats and are found only in specific life forms and seasons (Figs. 6A–6D).

Figure 6 Principal coordinates analysis (PCoA) plot showing the variation in species composition among sample groups based on Euclidean distance.

Each point represents a sample, and points closer together indicate higher similarity in species composition. (A) Bryophyte species, (B) bryophyte groups, (C) life forms, (D) rainfall. The axes PCoA1 and PCoA2 explain 31.37% and 20.55% of the total variation, respectively.

Figure 7 Principal Coordinates Analysis (PCoA) plot showing the variation in species composition among sample groups based on Euclidean distance.

Each point represents a sample, and points closer together indicate higher similarity in species composition. (A) Moss characteristics, the axes PCoA1 and PCoA2 explain 42.58% and 26.33% of the total variation, respectively. (B) Liverwort characteristics, the axes PCoA1 and PCoA2 explain 30.07% and 18.18% of the total variation, respectively.

At the species level, bryophytes exhibited a wide range of similarities in the bdelloid community, ranging from 0% to 67%, with a 41% similarity of community observed between moss and liverwort (Table S2). Notably, the moss species Frullania ericoides and the liverwort.

Taxithelium instratum were found to host the same bdelloid rotifer species (Table S3). Furthermore, species composition among life forms revealed the least resemblance, with the bdelloid community in cushion form being the most distinct, showing just 10% similarity to the turf form and 21% similarity to the mat form (Table S4). Additionally, H. flaviformis and M. multispinosa are typically found only in mat life forms (Fig. 6C). Additionally, the bdelloids community showed no significant seasonal differences, with similarities ranging from 44% to 55% (Table S5). The first two PCoA axes of moss characters accounted for a high proportion of the total variance (68.91%), with 42.58% explained by the first axis and 26.33% by the second axis (Fig. 7A). The similarity of species composition based on the characteristics of moss was 40% (Table S6). Four bdelloid rotifer species—Adineta sp.1, Habrotrocha sp., Philodina verrucosa, and Pleuretra sp.—tend to be specifically associated with moss that has leaves that curl when dry, while M. multispinosa and Macrotrachela sp. are more commonly associated with moss with leaves that do not curl when dry (Fig. 7A). Moreover, the first two PCoA axes of liverworts characters accounted for a high proportion of the total variance (48.25%), with 30.07% explained by the first axis and 18.18% by the second axis (Fig. 7B). Similarity in species composition based on liverwort characteristics was 57% (Table S6). Among these, Didymodactylos sp., Habrotrocha cf. alacris, H. bidens, M. multispinosa, and P. verrucosa tend to be found exclusively in liverworts with large lobules (Fig. 7B).

Ind indices and Habitat preference degree

All bdelloid rotifer species exhibited indicator values for bryophyte groups (mosses and liverworts) of less than 40%, indicating that these species are generally low indicators of mosses and liverworts. However, Scepanotrocha simplex and Habrotrocha cf. alacris displayed moderate habitat preferences for the liverwort species Cheilolejeunea ceylanica and Schiffneriolejeunea tumida var. haskarliana, respectively (Table 5).

Table 5 IndVal value and habitat preferences of species found in bryophyte species.

Bdelloid rotifer species	Relative abundance	IndVal (%) in bryophyte group	IndVal (%) in bryophyte species	Bryophyte species preference	Preference degree in bryophyte species	
Mosses	Liverworts					
A. cf. glauca	0	0.35	3.33	33.33	Without preference	Low	
Adineta sp.	0.79	0	4.55	14.29	Without preference	Low	
Didymodactylos sp.	0	0.35	3.33	14.29	Without preference	Low	
H. angusticollis	26.19	4.55	22.90	38.41	Without preference	Low	
H. cf. alacris	0	0.35	3.33	50.00	Without preference	Moderate	
H. bidens	4.76	9.09	16.24	11.30	Without preference	Low	
H. flaviformis	0	1.40	10.00	27.70	Without preference	Low	
H. gracilis	0	3.5	6.67	29.41	Without preference	Low	
Habrotrocha sp.	1.59	0	4.55	33.33	Without preference	Low	
M. multispinosa	33.33	11.19	29.56	24.06	Without preference	Low	
M. papillosa	0	2.45	16.67	35.61	Without preference	Low	
M. pinnigera	2.38	19.58	2.95	32.74	Without preference	Low	
Macrotrachela sp.	1.59	0	4.55	33.33	Without preference	Low	
P. verrucosa	0.79	0.35	4.08	17.51	Without preference	Low	
Pleuretra sp.	1.59	8.04	10.57	27.65	Without preference	Low	
R. sordida	26.98	23.78	26.33	17.64	Without preference	Low	
S. simplex	0	15.03	16.67	55.17	Without preference	Moderate	

Discussion

Species diversity

The present results confirmed high diversity of bdelloid rotifers in limnoterrestrial habitats. Of which, 22 taxa found in the present study accounting for about 4% of bdelloid species worldwide (Segers, 2007; Jersabek & Leitner, 2024). Moreover, 14 new records have increased the number of bdelloid rotifers in Thailand from 16 to 30 species (Sa-Ardrit, Pholpunthin & Segers, 2013; Maiphae, 2017; Jaturapruek et al., 2018, 2021). In addition, Habrotrocha flaviformis, Philodina verrucosa, and Scepanotrocha simplex, which were recorded for the first time in the Oriental region, have a broader distribution range than other species. Moreover, most species found in this study were reported for the first time in liverworts, except for Habrotrocha angusticollis and Macrotrachela multispinosa (Donner, 1965). Macrotrachela multispinosa and Rotaria sordida, found in every region except Antarctica (Segers, 2007), were the most numerous and frequently encountered in the present bryophyte samples. In particular, R. sordida has been recognized as a successful anhydrobiotic species (Eyres et al., 2015), suggesting it is an effective disperser that may have colonized this area before spreading more widely. However, the degree of tolerance to environmental variables of each species can also explain its distribution and abundance (Ricci, 1998).

The results showed that more bdelloid species inhabit liverworts than mosses, possibly because liverworts offer a more suitable habitat for bdelloid rotifers. Liverworts often have a thinner and more delicate structure compared to mosses, which can provide more intricate and varied microhabitats for bdelloids. The surface of liverworts may possess specialized structures, such as underleaves, imbricate leaves, or lobules, that offer hiding places or protection for small organisms (Inuthai, 2007; Kraichak, 2012). There have been reports of high numbers of bdelloid rotifer species inhabiting lobules, whether the lobule was characterized as a sac (Puterbaugh, Skinner & Miller, 2004) or not (Glime, 2017a). Microlejeunea punctiformis, which has large lobules and small leaves but a higher number of leaves than other species, was found to maintain a high number of bdelloid rotifer species. Therefore, these characteristics might increase microhabitat diversity or complexity. However, Lejeunea adpressa, characterized by its small lobules and simple lobule shape, also supports a high number of bdelloid rotifer species. This suggests that other factors, such as phytochemicals, may play a significant role in determining habitat suitability (Puterbaugh, Skinner & Miller, 2004; Xie & Lou, 2009).

Additionally, while the complexity of the bryophyte is an important factor, moisture content likely contributes to the greater richness and abundance of individuals observed (Hirschfelder, Koste & Zucchi, 1993). Unfortunately, moisture content was not measured in the present study, making it a crucial consideration for future research. Furthermore, liverworts were frequently found in areas protected by trees, which helped slow water loss, whereas mosses were found in more open areas, increasing the risk of desiccation. Moreover, most liverworts found in the present study grew in a mat life form, which retains moisture well (Proctor, 1990). This moisture retention is crucial for many invertebrates that require high humidity levels to survive, especially in dry environments (Schwarz et al., 1993; Ricci & Fontaneto, 2009; Velasco-Castrillón et al., 2014a; Devetter et al., 2017). Additionally, it has been reported that liverworts tend to decompose more readily than mosses, releasing nutrients into the environment at a faster rate (Lang et al., 2009). This decomposition process supports a diverse community of microorganisms and detritivores, which in turn attract various invertebrates that feed on them or utilize them as a resource. Some invertebrates directly consume liverwort tissues or use them as a substrate for feeding and reproduction (Haines & Renwick, 2009). Liverworts may offer a richer source of food or organic matter compared to mosses in certain ecosystems. Further study on the effect of food availability in liverworts and mosses on bdelloid rotifers and other invertebrates is needed.

However, surprisingly, in periods with moderate and high rainfall, which are characterized by high humidity, fewer species were found. One possible explanation is that during the dry season, the availability of water and suitable habitats may be limited. Under these conditions, bryophytes can serve as refuges for bdelloid rotifers, offering protection from desiccation and potentially reducing competition with other organisms. Additionally, the reduced water volume and simpler community structure in bryophyte-associated microhabitats during the dry season may lower predation and parasite pressure on bdelloid rotifers, allowing for higher species diversity to be sustained (Wilson, 2011; Wilson & Sherman, 2013). Moreover, the observed patterns in low- rainfall tropical regions may be similar to those found in other environments (Fontaneto, Iakovenko & De Smet, 2015). However, this hypothesis should be further tested in tropical ecosystems. Another possible reason is that bdelloid rotifers have unique adaptations, such as desiccation tolerance and dormancy strategies, that allow them to survive the harsh environmental conditions characteristic of the dry season (Caprioli & Ricci, 2001; Hespeels et al., 2023). For example, Habrotrocha gracilis, found during the low rainfall period, can secrete mucus to cover its body or combine with detritus to form a nest (Donner, 1965; Song & Kim, 2000). This trait, commonly found in this genus, may help Habrotrocha gracilis survive in harsh environments (Kutikova, 2003). Additionally, Philodina verrucosa has a thick and rough integument (Donner, 1965), a feature often seen in species that live in dry environments (Kutikova, 2003). These adaptations enable rotifers to persist within bryophytes despite fluctuating moisture levels and other environmental stressors, thereby contributing to sustained species diversity. In addition, the seasonal dynamics of bryophyte-associated habitats, influenced by moisture availability and temperature fluctuations, may create temporal niches that favour different stages of bdelloid rotifers (Ricci, Pagani & Bolzern, 1989). Consequently, this temporal variation can enhance species diversity by supporting a succession of bdelloid rotifer species adapted to varying ecological conditions throughout the dry season.

Habitat preference

The similarities in species composition among bryophyte groups, life forms, and bryophyte characteristics are high. These results indicate a limited level of habitat specialization of bdelloid rotifers in tropical limnoterrestrial environments, which may be due to several factors, including habitat characteristics and the species’ capacities for surviving desiccation, achieving long-term colonization, and being dispersed by wind and raindrops (Burger, 1948; Örstan, 1998; Fontaneto et al., 2007; Bielańska-Grajner, Mieczan & Cieplok, 2017). Bryophytes provide a moist environment, which is crucial for bdelloid rotifers and other microorganisms, as they rely on water films to move and feed (Hingley, 1993). Moreover, the dense structure of bryophytes offers protection against environmental stressors such as UV radiation and desiccation, while bdelloid rotifers contribute to nutrient cycling within the bryophyte ecosystem by feeding on detritus, bacteria, and other microorganisms (Glime, 2017b). However, although variations in species composition across variables were low, the results revealed that the cushion form exhibited the lowest similarity with species composition in other forms. This may be because this form has a more complex structure that supports different species compared to the more uniform structures of mat and turf. Additionally, cushions may maintain more varied moisture levels, microclimates, and food resources, attracting specialized species and leading to more distinct bdelloid communities. Different bryophyte species exhibit unique life forms and densities that influence moisture retention, potentially impacting species composition, particularly in mat form that retain water effectively (Glime, 2017c). In this study, bdelloid rotifer species found in Frullania ericoides and Taxithelium instratum were widely distributed, showing high abundance. Structural adaptations in bryophytes, such as curled leaves in mosses or large lobules in liverworts, may increase surface area, reduce water loss rates, and provide temporary refuge for small organisms (Kutikova, 2003). However, no specificity was observed between bdelloid rotifer species and bryophytes. Therefore, further research is needed to better understand how these factors influence the species composition of bdelloid rotifers.

Conclusions

This study confirms a high diversity of bdelloid rotifers in limnoterrestrial habitats. Species composition shows low similarity across different bryophyte species, forms, characteristics, and seasons, with no consistent relationship observed between bdelloid and bryophyte species, except for a few species. These findings suggest a low degree of habitat specialization within the bdelloid rotifer community in tropical limnoterrestrial environments.

Supplemental Information

Supplemental Information 1 All data used for the analyses.

Supplemental Information 2 Adineta cf. glauca.

(A) creeping, dorsal view; (B) creeping trunk, dorsal view; (C) foot, spurs and toes, lateral view. Adineta vaga (D) creeping, dorsal view. Habrotrocha cf. alacris (E) creeping, dorsal view; (F) creeping trunk and trophi, dorsal view. Habrotrocha angusticollis (G) contracting, dorsal view. Habrotrocha bidens (H) creeping, dorsal view; (I) creeping trunk and trophi, dorsal view; (J) foot and spurs, dorsal view. Habrotrocha cf. brocklehursti (K) creeping, dorsal view; (L) creeping trunk and trophi, dorsal view; (M) foot and spurs, dorsal view. (scale bars: B-C, F, I-J, K-L = 10 µm; A, D-E, G-H, K = 50 µm).

Supplemental Information 3 Habrotrocha flaviformis .

(A) creeping, lateral view; (B) feeding, lateral view. Habrotrocha gracilis (C) creeping, lateral view; (D) trunk and foot, lateral view. Macrotrachela multispinosa (E) creeping, dorsal view; (F) feeding, dorsal view. Macrotrachela papillosa (G) creeping, dorsal view; (H) trunk and foot, dorsal view. Macrotrachela pinnigera (I) contracting, dorsal view. Macrotrachela cf. plicata (J) creeping, dorsal view; (K) creeping trunk, lateral view. (scale bars: A-K = 50 µm).

Supplemental Information 4 Philodina rugosa .

(A) contracting, dorsal view; (B) creeping trunk, dorsal view; (C) foot and spurs, lateral view; Philodina verrucosa (D) contracting, dorsal view. Rotaria sordida (E) creeping, dorsal view; (F) head, dorsal view; (G) foot, dorsal view. Scepanotrocha simplex (H) creeping, dorsal view; (I) feeding, dorsal view. (scale bars: C, G = 10 µm; A-B, D-F, H-I = 50 µm).

We would like to thank the editor and reviewers for their time and effort in reviewing our manuscript. We sincerely appreciate the valuable comments and suggestions, which have greatly contributed to improving its quality. We are also grateful to Dr. Patsakorn Tiwutanon and Assoc. Prof. Dr. Ekaphan Kraichak for their assistance with bryophyte identification.

Additional Information and Declarations

Competing Interests

Author Contributions

Field Study Permissions

Data Availability

The authors declare that they have no competing interests.

Sittikron Jattupan conceived and designed the experiments, performed the experiments, analyzed the data, prepared figures and/or tables, authored or reviewed drafts of the article, and approved the final draft.

Rapeepan Jaturapruek performed the experiments, analyzed the data, prepared figures and/or tables, authored or reviewed drafts of the article, and approved the final draft.

Phannee Sa-ardrit performed the experiments, analyzed the data, prepared figures and/or tables, authored or reviewed drafts of the article, and approved the final draft.

Janejaree Inuthai performed the experiments, authored or reviewed drafts of the article, and approved the final draft.

Chatchai Ngernsaengsaruay performed the experiments, prepared figures and/or tables, authored or reviewed drafts of the article, and approved the final draft.

Supiyanit Maiphae conceived and designed the experiments, performed the experiments, analyzed the data, prepared figures and/or tables, authored or reviewed drafts of the article, and approved the final draft.

The following information was supplied relating to field study approvals (i.e., approving body and any reference numbers):

This research was approved by the Institutional Animal Care and Use Committee, Kasetsart University, Thailand (approval no. ACKU66-SCI-018).

The following information was supplied regarding data availability:

The raw data are available in the Supplemental Files.

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
