# Peer review of "Diversity and habitat preferences of bdelloid rotifers in mosses and liverworts from beach forest along sand dunes in Thailand"

_PeerJ, doi:10.7717/peerj.18721_

## Round 0.1 · original submission · Major Revisions

I agree with the reviewers that the manuscript needs considerable work to be published, but that you have an interesting study that has great potential. Please carefully read the suggestions of the two reviewers as well as the annotated manuscript I include.

An important issue is the analyses. I think you spend too much time discussing species richness when we know this index is a poor measure of community structure or processes. I think you could include principle coordinate analysis and other measures of community similarities to better explain the distribution of the rotifers. For example, by showing measures of similarity of rotifers between liverworts and bryophytes, or seasons, or locations, you can get a handle on HOW rotifer distributions might be determined. You start off your introduction by talking about how small animals are distributed, which implies that a big part of your idea should be to explain that. Richness indexes do not do that, but distance measures easily could.

If you used PCoA, you can provide more informative figures that would serve your purposes better than the cluster analyses (or, maybe in combination). Considering the community-structure nature of your study, I think focussing on that kind of analysis rather than using richness indexes would make a much stronger study.

Also note my comments on writing style and be sure to think about how to apply those ideas throughout the text.

Reviewer 1 ·

Basic reporting

Please, see attached file

Experimental design

Please, see attached file

Validity of the findings

Please, see attached file

Additional comments

Please, see attached file

Annotated reviews are not available for download in order to protect the identity of reviewers who chose to remain anonymous.

Reviewer 2 ·

Basic reporting

This research is significant as it offers a distinctive perspective on the variety and habitat preferences of bdelloid rotifers residing in bryophytes (mosses and liverworts) within beach forests along sand dunes in Thailand. The study employs a rigorous methodology, is well-referenced, and includes a clear results table.
The English is generally acceptable; however, the manuscript contains several significant grammatical errors that constrain its potential for publication in its current form, as well as sentences with atypical phrasing.

The site requires comprehensive data on the forest area and altitude. Do the altitudes remain consistent across all sampled bryophytes, or do they show variation?

The methodology requires additional information on climate data, particularly regarding rainfall, to properly present the different seasons mentioned in the text (low rainy season, mid rainy season, high rainy season).

The results are clearly presented; however, additional development is necessary to improve their originality and relevance to the journal's readership.

Some more specific suggestions are detailed below.

In lines 61-62, it is preferable to state: 'In addition, certain small organisms, such as rotifers, possess efficient resting stages that allow them to endure prolonged periods of dormancy, and their asexual...

Line 94: "In the Introduction, the presentation of the research topic should be refined to highlight the significance of the Bang Berd Beach Forest and the diversity of bryophytes, followed by a clear statement of the study's objectives."

Line 101: "The objectives of the study should be clearly defined, with a focus on examining the distribution patterns of bdelloid rotifers in relation to bryophyte species and their characteristics."


In Data analysis
Line 144
For analyzing species composition and habitat preferences, various analytical methods may be applicable, depending on the specific objectives of the study and the nature of the available data. In addition to Cluster Analysis, the use of other established methods is recommended. Some suggested approaches include:

Principal Component Analysis (PCA): A multivariate technique used to reduce data dimensionality and identify underlying patterns, facilitating the exploration of relationships between habitat types and species composition.
Redundancy Analysis (RDA): A constrained ordination method used to assess the relationships between species composition and a set of explanatory variables, such as environmental gradients.
Chi-square tests and Analysis of Variance (ANOVA/MANOVA): Statistical tests employed to detect significant differences in species composition or habitat preferences across different groups or study sites.

Experimental design

See above

Validity of the findings

See above

Additional comments

See above

---

## Round 0.2 · Major Revisions

While this version is somewhat improved, I do not think it is improved enough to send to the reviewers again. Please note that I am enclosing the annotated copy with my remarks and observations. I must ask you to also go back to my original annotated manuscript that I sent the last time, because there are several issues that you did not address.

My main concern is with your analyses. For example, according to your table 5 in which you compared richness, diversity and evenness between bryophytes you showed that all bryophytes have few rotifers so these indices are not the best way to make these comparisons. You have too many bryophytes and too few rotifers for this comparison to make sense. I previously recommended, and continue to recommend, that you used PCoA or some other kind of community analysis rather than simple indices.

Your PCA analyses are entirely opaque. You write that you log transformed variables but you never state what the numerical variables are that can be transformed. In figures 6 through 11 (that I previously recommended you combine into fewer figures) you never interpret what the axes mean - what variables comprise them. And, while they do show some patterns, none explains most of the variance, meaning they are all relatively unimportant. The axes of the figures should have numbers associated with them, but you simply put Axis 1 and Axis 2.

You have too many tables that are too complicated to be particularly useful. I recommend that you find a way to simplify them, or keep them as supplemental information.

The first sentence of your conclusion (2nd, if you keep the one I removed) says "The results of the present study on bdelloid rotifers in bryophytes confirm their limited distribution across different microhabitats and seasons" but I'm not convinced. That is, your statistics are questionable, the percentage explained by PCA is consistently small, and the figures show a lot of overlap. The fact that you have several species of bryophytes with no or few rotifers is simply because you have so few rotifers in general. It's not surprising that they don't always appear in a sample. In any case, I strongly recommend that you use different statistics than PCA and that you explain your variables better. PCA requires numerical variables - what are yours.

Because these suggestions of mine also carry over from the first draft, I am not sending this back to the reviewers because I do not see big and important enough changes to warrant the efforts of the reviewers. Please reread all the first reviews as you revise this version.

---

## Round 0.3 · accepted · Accept

I just have a few minor comments. The numbers over the bars in Figure 4 are unnecessary. Some of the colors in figures 6 and 7 do not work well together visually. Perhaps you can add an outline with a line-type that varies by grouping. The text size in those figures is also fairly small, so you might enlarge it to make it more legible when the figures are reduced to publication size.

Reviewer 1 ·

Basic reporting

The study is well-structured and clearly presented
Thorough background was provided
References seem to be valid. Results are relevant.
English language was improved.


All my comments and suggestions relating to reporting were considered. I see that manuscript was improved significantly. I much appreciate notes on identification of bdelloids

The only thing that still may be improved is the abstract. Since the authors used new statistical methods (PCoA), some results of these analyses can be briefly mentioned in the abstract.

Experimental design

The experimental design is consistent with the issues raised in the paper.

Validity of the findings

All findings are valid

Additional comments

no comments

Reviewer 2 ·

Basic reporting

The revised manuscript meets high standards of clarity and scientific communication. The language is concise, the references are relevant, and the introduction provides clear context. Visual elements, including figures and tables, effectively support the findings. Overall, the basic reporting is well-aligned with the journal's standards.

Experimental design

The revisions have clarified the presentation of results, particularly through the improved explanation and use of PCoA, which provides better insight into the relationships between variables. The methodology is now more clearly articulated, enhancing reproducibility and understanding. Overall, the design is rigorous and significantly contributes to the clarity and scientific value of the study.

Validity of the findings

ok

Additional comments

I commend you for the meticulous effort invested in revising your manuscript. You have successfully addressed all the feedback and incorporated the necessary corrections, significantly enhancing the overall quality and scientific value of your work.